# Code Generation as a Dual Task of Code Summarization

**Bolin Wei[†], Ge Li[†*], Xin Xia[‡], Zhiyi Fu[†], Zhi Jin[†*]**
[†]Key Laboratory of High Confidence Software Technologies (Peking University)
Ministry of Education, China; Software Institute, Peking University, China
[‡]Faculty of Information Technology, Monash University, Australia
[†]`bolin.wbl@gmail.com`  [†]`{lige,ypfzy,zhijin}@pku.edu.cn`
[‡]`xin.xia@monash.edu`

## Abstract

Code summarization (CS) and code generation (CG) are two crucial tasks in the field of automatic software development. Various neural network-based approaches are proposed to solve these two tasks separately. However, there exists a specific intuitive correlation between CS and CG, which has not been exploited in previous work. In this paper, we apply the relations between two tasks to improve the performance of both tasks. In other words, exploiting the duality between the two tasks, we propose a dual training framework to train the two tasks simultaneously. In this framework, we consider the dualities on probability and attention weights, and design corresponding regularization terms to constrain the duality. We evaluate our approach on two datasets collected from GitHub, and experimental results show that our dual framework can improve the performance of CS and CG tasks over baselines.

## 1   Introduction

Code summarization (CS) is a task that generates comment for a piece of the source code, whereas code generation (CG) aims to generate code based on natural language intent, e.g., description of requirements. Code comments, a form of natural language description, provide a clear understanding for users, and are very useful for software maintenance [de Souza et al., 2005]. On the other hand, CG is an indispensable process in which programmers write code to implement specific intents [Balzer, 1985]. Proper comments and correct code can massively improve programmers' productivity and enhance software quality. However, generating the correct code or comments is costly, time-consuming, and error-prone. Therefore, carrying out CS and CG automatically becomes greatly important for software development.

Recently, many researchers, inspired by an encoder-decoder framework [Cho et al., 2014], applied neural networks to solve these two tasks independently. For CS, the encoder uses a neural network to represent source code as a real-valued vector, and the decoder uses another neural network to generate comments word by word. The main difference among previous studies is the way to encode source code. Specifically, Iyer et al. [2016] and Hu et al. [2018a,b] modeled source code as a sequence of tokens, composed by original code tokens or abstract syntax tree (AST) nodes obtained by traversing ASTs in a certain order. On the other hand, Wan et al. [2018] treated a code snippet as an AST. It is worth noting that these previous models have both introduced the attention mechanism [Bahdanau et al., 2015] to learn the alignment between the code and the comment.

---

[*]Corresponding authors.

For CG with a general-purpose programming language, the encoder-decoder framework was first applied by Ling et al. [2016], where the encoder took a natural language description as input, and the decoder generated the source code. In addition, some researchers used auxiliary information to improve the performance of their models, such as grammar rules [Yin and Neubig, 2017, Rabinovich et al., 2017, Sun et al., 2018] and structural descriptions [Ling et al., 2016]. These methods all applied the attention mechanism as in CS task. Overall, previous studies on CS and CG were independent of each other. None of the previous studies before have considered the relations between the two tasks or exploited the relations to improve each other.

Intuitively, CS and CG are related to each other, i.e., the input of CS is the output of CG, and vice versa. We refer to this relation as duality, which provides some utilizable constraints to train the two tasks. Specifically, from the perspective of probability, given a piece of source code and a corresponding comment, there exists a pair of inverse conditional probabilities between them, bound by their common joint probability. From the perspective of the model structure, both tasks take the encoder-decoder framework, with source code and comment as both input and output. We conjecture that the attention weights from the two models should be as similar as possible because they both reflect the similarity between the token at one end and the token at the other end. Besides, the CS and the CG model require similar abilities in understanding natural language and source code. Thus, we argue that the joint training of the two models can improve the performance of both models, especially when we add some constraints to this duality.

In this paper, we design a dual learning framework to train a CS and a CG model simultaneously to exploit the duality of them. Besides applying a probabilistic correlation as a regularization term [Xia et al., 2017] in the loss function, we design a novel constraint about attention mechanism to strengthen the duality. We conduct our experiments on Java and Python projects collected from GitHub used by previous work [Hu et al., 2018a,b, Wan et al., 2018]. Experimental results show that jointly training two models can help the CS model and CG model outperform the state-of-the-art models.

The contributions of our work are shown as follows:

- To our best knowledge, it is the first time we propose a joint model for automated CS and CG. We unprecedentedly attempt to treat CS and CG as dual tasks and apply a dual framework to boost each other.

- We adopt a probabilistic correlation between CS and CG as a regularization term in loss function and design a novel constraint to guarantee the similarity of attention weights from two models in the training process.

## 2 Related Work

Code summarization (CS), as an essential part of the software development cycle, has attracted a lot of recent attention. With the development of deep learning, neural networks are applied in this task successfully. Besides Allamanis et al. [2016] using a CNN to generate short and name-like summaries, most of the related work followed the encoder-decoder framework. Iyer et al. [2016] used an RNN with attention mechanism as a decoder. Hu et al. [2018a] introduced a machine translation model to generate summaries for Java methods given the serialized AST. Then they adopted a transfer learning method to utilize API information to generate code summarization [Hu et al., 2018b]. Wan et al. [2018] designed a tree RNN, which leverages code structure information, and applied a reinforcement learning framework to build the model. Note that since the tree-based module must take a tree (e.g., AST) as input and different training samples have different tree structures, it is hard to carry out batch processing in the implementation.

As a fundamental part of software development, code generation (CG) is a popular topic among the field of software engineering as well. Recently, more and more researchers apply neural networks for general-purpose CG. Ling et al. [2016] introduced a sequence-to-sequence model to generate code from natural language and structured specifications. Yin and Neubig [2017] and Rabinovich et al. [2017] both combined the grammar rules with the decoder and improved code generation performance. Sun et al. [2018] argued that traditional RNN might not handle the long dependency problem, and they proposed a grammar-based structural CNN. Different from the aforementioned methods, we argue that a simple CG model could boost the CS model in our dual framework, and the CG model can also benefit from the dual training process.

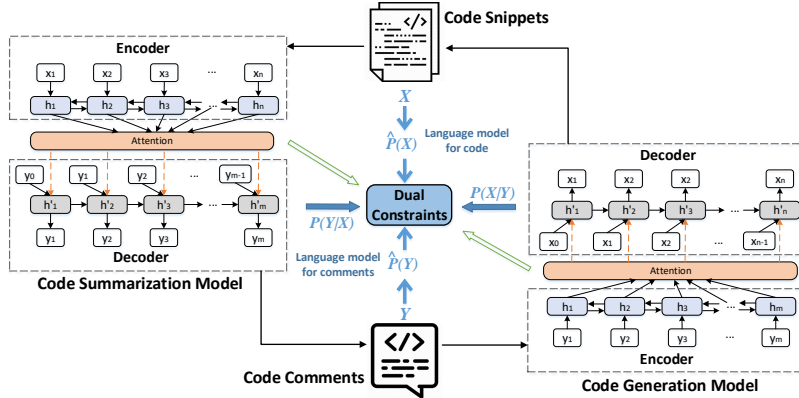

Figure 1: The overall dual training framework. A CS model and a CG model are trained jointly in the framework.

Dual learning, proposed by He et al. [2016], is a reinforcement training process that jointly trains a primal task and its dual task. Xia et al. [2017] considered it as a way of supervised learning and designed a probabilistic regularization term to constrain the duality, which has been applied successfully in machine translation, sentiment classification, and image recognition. Tang et al. [2017] treated question answering and question generation as a dual task and proved the effectiveness of dual supervised learning. Li et al. [2018] applied a dual framework to improve the performance of visual question answering with the help of visual question generation. Xiao et al. [2018] introduced a dual framework to jointly train a question answering model and a question generation model in machine reading comprehension. To the best of our knowledge, we are the first to propose a dual learning framework in CS and CG, and leverage the duality between them. Furthermore, we design a new dual constraint about attention weights to strengthen the correlation between the two tasks.

## 3   Proposed Approach

Our training framework, illustrated in Figure 1, consists of three parts: a CS model, a CG model, and dual constraints. The CS model aims to translate the source code to a comment, while the CG model maps natural language description to a source code snippet. Dual constraints are used by adding regularization terms in the loss function to constrain the duality between two models. We first formulate the tasks of CS and CG and then describe the details of our framework.

We denote a set of source code snippets as $\mathcal{X} = \{x^{(i)}\}$ where $x^{(i)} = \{x_1^{(i)}, ..., x_n^{(i)}\}$ denotes the token sequence of a code snippet. Each code snippet $x^{(i)}$ has a corresponding natural language comment as $y^{(i)}$ where $y^{(i)} = \{y_1^{(i)}, ..., y_m^{(i)}\} \in \mathcal{Y}$. Thus, the CS model learns a mapping $f_{xy}$ from $\mathcal{X}$ to $\mathcal{Y}$ and the CG model learns a reverse mapping $f_{yx}$ from $\mathcal{Y}$ to $\mathcal{X}$. Different from previous work, we regard the two tasks as a dual learning problem and train them jointly.

### 3.1   Code Summarization Model

The CS model takes a code snippet $x^{(i)}$ as input to generate a comment. Considering the efficiency, we choose the sequence-to-sequence (Seq2Seq) neural network with an attention mechanism as our model. The model contains two parts: an encoder and a decoder.

The encoder first maps the token of a source code snippet into a word embedding. Words which do not appear in the vocabulary are defined as *unknown*. Then to leverage the contextual information, we use a bidirectional LSTM to process the sequence of the word embeddings. We concatenate hidden states of the time step $i$ from two directions as the representation of the $i$-th token $h_i$ in source code.

The decoder is another LSTM with an attention mechanism between encoder and decoder, which generates a word $y_t$ based on the representation of whole source code snippet and the previous words

in the comment. This process is formulated as

$$P(y|x) = \prod_{t=1}^{m} P(y_t|y_{<t}, x) \tag{1}$$

The last hidden state from the encoder is used to initialize the hidden state of the decoder which computes the hidden state as $h_t' = \text{LSTM}(y_{t-1}', h_{t-1}')$, where $y_{t-1}'$ is defined as the concatenation of the embedding of $y_{t-1}$ and the attention vector $a_{t-1}$. $h_t'$ is used to compute the attention weights as

$$\widetilde{\alpha}_{ti} = h_t'^{\top} W h_i \tag{2}$$

$$\alpha_{ti} = \frac{\exp\{\widetilde{\alpha}_{ti}\}}{\sum_j \exp\{\widetilde{\alpha}_{tj}\}} \tag{3}$$

where $W$ is a group of trainable weights. The attention score $\alpha_{ti}$ measures the similarity between the token of comment $y_t$ and the token of code snippet $x_i$. We compute the context vector $c_t$ as $c_t = \sum_j \alpha_{tj} h_j$. Then $c_t$ and $h_t'$ are fed into a feed-forward neural network to get the attention vector $a_t$ which is fed into a softmax layer to get the prediction of the token $y_t$ in the comment. We use negative log-likelihood as the training objective. The loss function of a training sample is

$$l_{xy} = -\frac{1}{m} \sum_{t=1}^{m} \log P(y_t|y_{<t}, x) \tag{4}$$

## 3.2 Code Generation Model

Unlike work depending on the grammar rules, our CG model, which can be regarded as the inverse CS task, predicts the code snippet only based on natural language description. We use the same structure as the CS model, i.e., a Seq2Seq neural network. The encoder is a bidirectional LSTM, and the decoder is another LSTM with an attention mechanism. Different from the CS model, the decoder in the CG model learns a conditional probability as

$$P(x|y) = \prod_{t=1}^{n} P(x_t|x_{<t}, y) \tag{5}$$

And the training objective is formulated as

$$l_{yx} = -\frac{1}{n} \sum_{t=1}^{n} \log P(x_t|x_{<t}, y) \tag{6}$$

Since the source code contains lots of identifiers, the vocabulary size of the source code is usually much larger than that of comments, which results in more parameters in the output layer of the CG model than in the output layer of the CS model.

## 3.3 Dual Training Framework

The dual training framework includes three components: a CS model, a CG model and two dual regularization terms to constrain the duality of the two models, which are enlightened by the probabilistic correlation and the symmetry of attention weights between two models.

Given a pair of $\langle x, y \rangle$, the CS model and the CG model have probabilistic correlation, because they are both connected to the joint probability $P(x, y)$ which can be computed in two equivalent ways.

$$P(x, y) = P(x)P(y|x) = P(y)P(x|y) \tag{7}$$

Since the CS model, parameterized by $\theta_{xy}$, is built to learn the conditional probability $P(y|x; \theta_{xy})$, and the CG model, parameterized by $\theta_{yx}$, is built to learn the conditional probability $P(x|y; \theta_{yx})$, we can jointly train these two models by minimizing their loss functions subject to the constraint of Eqn. 7. We build the constraint of Eqn. 7 to a penalty term by using the method of Lagrange multipliers, and thus our regularization term is

$$l_{dual} = [\log \hat{P}(x) + \log P(y|x; \theta_{xy}) - \log \hat{P}(y) - \log P(x|y; \theta_{yx})]^2 \tag{8}$$

---
**Algorithm 1** Algorithm Description

---

**Input:** Language models $\hat{P}(x)$ and $\hat{P}(y)$ for any $x \in \mathcal{X}$ and $y \in \mathcal{Y}$; hyper parameters $\lambda_{dual1}$, $\lambda_{dual2}$, $\lambda_{att1}$ and $\lambda_{att2}$; optimizers $opt1$ and $opt2$

**repeat**
    Get a minibatch of $k$ pairs $\langle (x_i, y_i) \rangle_{i=1}^k$;
    Calculate the gradients for $\theta_{xy}$ and $\theta_{yx}$.

$$G_{xy} = \nabla_{\theta_{xy}}(1/k)\sum_{i=1}^{k}[l_{xy}(f_{xy}(x_i;\theta_{xy}),y_i) + \lambda_{dual1}l_{dual}(x_i,y_i;\theta_{xy},\theta_{yx}) + \lambda_{att1}l_{att}(x_i,y_i;\theta_{xy},\theta_{yx})];$$

$$G_{yx} = \nabla_{\theta_{yx}}(1/k)\sum_{i=1}^{k}[l_{yx}(f_{yx}(y_i;\theta_{yx}),x_i) + \lambda_{dual2}l_{dual}(x_i,y_i;\theta_{xy},\theta_{yx}) + \lambda_{att2}l_{att}(x_i,y_i;\theta_{xy},\theta_{yx})];$$

    Update $\theta_{xy}$ and $\theta_{yx}$
    $\theta_{xy} \leftarrow opt_1(\theta_{xy}, G_{xy}), \theta_{yx} \leftarrow opt_2(\theta_{yx}, G_{yx})$
**until** models converged

---

where $\hat{P}(x)$ and $\hat{P}(y)$ are marginal distributions, which can be modeled by their language models, respectively. By minimizing this dual loss function, the probabilistic connection between the two models could be strengthened, which is helpful to the training process.

Naturally, structures of the CS model and CG model have symmetries, meaning that the output of one model is the input of the other model, and vice versa. Thus, we can introduce this property to the dual training process. In this paper, we focus on the symmetry of attention weights and argue that the alignment between tokens in the source code snippet and tokens in the comment has symmetry, which could be measured by attention weights. Specifically, given the comment "*find the position of a character inside a string*" and the corresponding source code "*string . find ( character )*", no matter how the generating direction goes, the word "*find*" in the comment should always be aligned to the same token "*find*" in the source code. Hence, we design a regularization term to leverage this duality.

The matrices of attention weights before normalization in the CS and the CG model, easily obtained by Eqn. 2, are denoted as $A_{xy} \in \mathbb{R}^{n \times m}$ and $A_{yx} \in \mathbb{R}^{m \times n}$. The element $\alpha_{ij}$ in $A_{xy}$ and the element $\alpha_{ji}$ in $A_{yx}$ both measure the similarity between the $i$-th token in the source code with the $j$-th token in the comment. For the $i$-th token in the source code, we obtain its attention weights $b_i$, a probability distribution, from the CS model by $b_i = \text{softmax}(A_{xy}^i)$, where $A_{xy}^i$ is the $i$-th row vector of $A_{xy}$. Then the attention weights from CG model is $b_i' = \text{softmax}(A_{yx}^i)$, where $A_{yx}^i$ is the $i$-th column vector of $A_{yx}$. Finally, we apply the Jensen–Shannon divergence [Fuglede and Topsoe, 2004], a symmetric measurement of similarity between two probability distributions, to constrain the distance between these two attention weights. Thus, the penalty term $l_1$ for tokens in the source code is

$$l_1 = \frac{1}{2n}\sum_{i=1}^{n}[D_{KL}(b_i \| \frac{b_i + b_i'}{2}) + D_{KL}(b_i' \| \frac{b_i + b_i'}{2})] \tag{9}$$

where $D_{KL}$ is the Kullback–Leibler divergence, defined as $D_{KL}(p\|q) = \sum_x p(x)log\frac{p(x)}{q(x)}$, which measures how one probability distribution $p$ diverges from the other probability distribution $q$. Likewise, we can obtain the penalty term $l_2$ for tokens in comments in the same manner. Consequently, the final regularization term about attention weights $l_{att}$ is the sum of $l_1$ and $l_2$. Moreover, we consider that the attention weights from one model could be regarded as the soft label of the other model. The overall algorithm is described in Algorithm 1, and the complexity of our model is the same as the Seq2Seq neural network. Our implementation is based on PyTorch.[2]

## 4 Experiments

### 4.1 Datasets

We conduct our CS and CG experiments on two datasets, including a Java dataset [Hu et al., 2018b] and a Python dataset [Wan et al., 2018]. The statistics of the two datasets are shown in Table 1.

Table 1: Statistics of datasets

| Dataset | Java | Python |
|---|---|---|
| Train | 69,708 | 55,538 |
| Validation | 8,714 | 18,505 |
| Test | 8,714 | 18,502 |
| Avg. tokens in comment | 17.7 | 9.49 |
| Avg. tokens in code | 98.8 | 35.6 |

The Java dataset, containing Java methods extracted from 2015 to 2016 Java projects, is collected from GitHub[3]. We process the dataset following Hu et al. [2018b]. The first sentence of Javadoc is extracted as the natural language description, which describes the functionality of the Java method. Each data sample is organized as a pair of ⟨method, comment⟩.

For our language model applied to Java dataset, we can use various large scale monolingual corpus to pretrain. In this paper, we use the Java projects from 2009 to 2014 on GitHub [Hu et al., 2018a] as our dataset. We separate the original datasets into a code-only dataset and a comment-only dataset. The language model of source code takes the Java methods in the code-only dataset as input, whereas the language model of comment takes the comments in the comment-only dataset as input. Each dataset is split into training, test and validation sets by 8:1:1.

The original Python dataset is collected by Barone and Sennrich [2017], consisting of about 110K parallel samples and about 160K code-only samples. The parallel corpus is used to evaluate CS task and CG task. Since Wan et al. [2018] used the dataset in their experiments, we follow their approach to process this dataset.

For the language model of source code in Python, we use the code-only samples in the original dataset to pretrain. We divide the samples into training, test, and validation sets by 8:1:1. However, for the language model of comments in Python, we are not able to find enough monolingual corpus to pretrain. Considering the patterns of comments in Python dataset and in Java dataset are very similar, we use the comment-only corpus from Java dataset as an alternative. Experimental results turn out that the language model pretrained in this way is beneficial to our dual training.

## 4.2 Hyperparameters

We set the token embeddings and LSTM states both to 512 dimensions for the CS model and set the LSTM states to 256 dimensions for the CG model to fit GPU memory. Afterward, to initialize the CS and the CG model in our dual training framework, we use warm-start CS and CG models, whose parameters are optimized by Adam [Kingma and Ba, 2015] with the initial learning rate of 0.002. Warm-starting means that we pretrained CS and CG models separately, then applied dual constraints to the two models for joint training, which can speed up the convergence process of joint training. The dropout rates of all models are set to 0.2 and mini-batch sizes of all models to 32. For dual learning process, we observe that the SGD is appropriate with initial learning rate 0.2. We halve the learning rate if the performance of the validation set decreases once and freeze the token embeddings if the performance decreases again. According to the performance of the validation set, the best dual model is selected after joint training 30 epochs in the experiments. The $\lambda_{dual1}$ and $\lambda_{dual2}$ are set to 0.001 and 0.01 respectively, and the $\lambda_{att1}$ and $\lambda_{att2}$ are set to 0.01 and 0.1. We use beam search in the inference process, whose size is set to 10. Furthermore, the vocabulary sizes of the code in Java and Python dataset are set to 30000 and 50000, and maximum lengths of code and comments in Java are set to 150 and 50 respectively.

Our language models for source code and comments both employ 3 LSTM layers. Token embeddings and LSTM states are both 300-dimensional. Batch size is set to 40 and dropout rate to 0.3. We apply gradient clipping to prevent gradients from becoming too large. The vocabulary consists of all words that have a minimum frequency of 3. Adam is chosen as our optimizer, and the initial learning rate is set to 0.002. During the dual training process, the parameters of language models are fixed. Language models are only used to calculate marginal probabilities $\hat{P}(x)$ and $\hat{P}(y)$ in Eqn. 8.

Table 2: The overall performance of our CS models compared with baselines

| Methods | Java | | | Python | | |
|---|---|---|---|---|---|---|
| | BLEU | METEOR | ROUGE-L | BLEU | METEOR | ROUGE-L |
| CODE-NN | 27.60 | 12.61 | 41.10 | 17.36 | 9.288 | 37.81 |
| DeepCom | 39.75 | 23.06 | 52.67 | 20.78 | 9.979 | 37.35 |
| Tree2Seq | 37.88 | 22.55 | 51.50 | 20.07 | 8.957 | 35.64 |
| RL+Hybrid2Seq | 38.22 | 22.75 | 51.91 | 19.28 | 9.752 | 39.34 |
| API+CODE | 41.31 | 23.73 | 52.25 | 15.36 | 8.571 | 33.65 |
| Basic Model | 41.01 | 23.26 | 51.64 | 20.47 | 10.38 | 38.77 |
| Dual Model | **42.39** | **25.77** | **53.61** | **21.80** | **11.14** | **39.45** |

Table 3: BLEU scores and percentage of valid code (PoV) on CG task

| Methods | Java | | Python | |
|---|---|---|---|---|
| | BLEU | PoV | BLEU | PoV |
| SEQ2TREE | 13.80 | 22.6% | 4.472 | 22.7% |
| Basic Model | 10.86 | 19.6% | 10.43 | 41.8% |
| Dual Model | **17.17** | **27.4%** | **12.09** | **51.9%** |

## 5 Experimental Results

**Metrics.** We evaluate the performance of CS task based on three metrics, BLEU [Papineni et al., 2002], METEOR [Banerjee and Lavie, 2005] and ROUGE-L [Lin, 2004]. These metrics all measure the quality of generated comments and can represent the human's judgment. BLEU is defined as the geometric mean of $n$-gram matching precision scores multiplied by a brevity penalty to prevent very short generated sentences. We choose sentence level BLEU as our metric as in Hu et al. [2018a,b]. METEOR combines unigram matching precision and recall scores using harmonic mean and employs synonym matching. ROUGE-L computes the length of longest common subsequence between generated sentence and reference and focuses on recall scores. For CG task, we choose BLEU as our performance metric because accuracies on both datasets are too low. Ling et al. [2016], Yin and Neubig [2017] and Sun et al. [2018] discussed the effectiveness of the BLEU in the CG task. They treated the BLEU as an appropriate proxy for measuring semantics and left exploring more sophisticated metrics as future work. Furthermore, to evaluate how much of the generated code is valid, we calculate the percentage of code that can be parsed into an AST.

**Baselines.** We compare our CS model's performance with the following five baselines.[4] It has been proved that the attention mechanism is very helpful to comment generation [Hu et al., 2018a], so all baselines introduced this module. **CODE-NN** [Iyer et al., 2016] uses token embeddings to encode source code and uses an LSTM to decode. To exploit the structural information, **DeepCom** [Hu et al., 2018a] takes a sequence of tokens as input, which is obtained through traversing the AST with a structure-based traversal method, while **Tree2Seq** [Eriguchi et al., 2016] directly uses a tree-based LSTM as an encoder. **RL+Hybrid2Seq** [Wan et al., 2018] is a model trained with reinforcement learning, whose encoder is the combination of an LSTM and an AST-based LSTM. We further compare with **API+CODE** [Hu et al., 2018b] without transferred API knowledge, which introduces API information when generating comments. The proportion of source code in Python having no APIs is about 20%; we set the prediction of test samples having no APIs to null. Except for Tree2Seq and RL+Hybrid2Seq (They used Word2Vec to pretrain their token embeddings), the token embeddings for other models are randomly initialized. For DeepCom, we use a bi-LSTM as the encoder to ensure the number of parameters is comparable to that of our model. For API+CODE, we set the embeddings and GRU states to 512 dimensions.

For CG model, we compare our dual model with the individually trained basic model, and also compare to **SEQ2TREE** [Dong and Lapata, 2016], which modeled the source code as a tree. Since our CG model only takes natural language description as input, to make a fair comparison, we do not compare with other models that take grammar rules [Yin and Neubig, 2017, Rabinovich et al., 2017, Sun et al., 2018] or structured specification [Ling et al., 2016] as additional input.

Table 4: Ablation study of different settings on CS task. Model (M) 1 is the basic model of independent training.

| M | Probabilistic Duality | Attention Duality | Java | | | Python | | |
|---|---|---|---|---|---|---|---|---|
| | | | BLEU | METEOR | ROUGE-L | BLEU | METEOR | ROUGE-L |
| 1 | - | - | 41.01 | 23.26 | 51.64 | 20.47 | 10.38 | 38.77 |
| 2 | ✓ | - | 41.73 | 25.54 | 53.60 | 21.66 | 10.81 | 38.83 |
| 3 | - | ✓ | 41.96 | 25.80 | 53.57 | 21.57 | 10.91 | 39.07 |
| 4 | ✓ | ✓ | 42.39 | 25.77 | 53.61 | 21.80 | 11.14 | 39.45 |

**Overall Results.** The overall performance of the CS model is shown in Table 2. Results show that our dual model obviously outperforms all the baselines on three metrics at the same time. To test whether the improvements of our dual model over baselines are statistically significant, we applied the Wilcoxon Rank Sum test (WRST) [Wilcoxon, 2006], and all the p-values are less than 0.01, indicating significant increases. We also used Cliff's Delta [Cliff, 1996] to measure the effect size, and the values are non-negligible. From results of baseline models, we can see that our independently trained basic model is simple and effective, yet is still inferior to the dual model, showing the effectiveness of joint training process. Compared to the sequence-based models, DeepCom and our basic model, the tree-based models (Tree2Seq and RL+Hybrid2Seq) do not achieve strong improvements of BLEU and METEOR scores on the two datasets. We suppose the reason for this phenomenon is that after the source code is converted to AST, the number of nodes becomes very large, resulting in increased noise. According to statistics, the average token numbers of Java and Python datasets have been more than doubled after parsing. In this case, due to the method's handling of the custom identifier contained in the code, DeepCom has achieved good results. The results of API+CODE indicate that API knowledge in Java dataset is beneficial to the comment generation. Though we do not focus on integrating structural information and API knowledge in the current work, we will leverage them in future studies considering their potential for boosting performance.

Since CS and CG models are trained at the same time and the parameters of the two models are separate after the joint training, i.e., the two models solve their respective tasks separately after the joint training, the number of parameters of each dual model is the same as that of the basic model. The number of parameters for all models on Java CS task is as follows: CODE-NN 34M, DeepCom 53M, Tree2Seq 52M, RL+Hybrid2Seq 80M, API+CODE 80M, Basic model 53M and Dual model 53M. The relationship between the number of all models' parameters is consistent in Java and Python experiments on CS task.

The results of CG model are shown in Table 3. Although the BLEU score is very low, dual training can still improve the performance of individually trained basic model, proving the effectiveness of dual training. It is observable that SEQ2TREE's performance is better than our basic model's on Java dataset, which demonstrates the ability of SEQ2TREE to leverage code hierarchies. However, its performance is far worse than our basic model's on Python dataset. The reason is that SEQ2TREE builds up tree structure according to brackets contained in the code, while hierarchical levels in Python code are segmented by line breaks and indents. Besides, dual training can also increase the percentage of valid code on both Java and Python datasets.

**Component Analysis.** We verified the role of dual regularization terms in joint training on CS task. The experimental results are shown in Table 4. Model 1 is the basic model of independent training. We can see that the introduction of CG model to train models jointly and constrain the duality between the two models can improve the performance of the CS model, indicating the utilizability of the relation between the two tasks. Specifically, we find that applying a regularization term on attention is slightly more effective than applying a regularization term on probability to improve model's performance. This is because, intuitively, the attention regularization term has a more powerful and more explicit constraint than the probability regularization term. Naturally, combining the two will further enhance our CS model's performance. To test whether the improvements of two constraints over one constraint on BLEU are significant, we applied the WRST, and all the p-values are less than 0.05, indicating significant increases. Cliff's Delta values also show non-negligible improvements.

In order to better understand the role of regularization terms, we also conduct experiments on the Python dataset to observe changes in regularization terms. In particular, after exerting the two constraints, the value of regularization term on probability is reduced from 129.1 to 125.9, and the

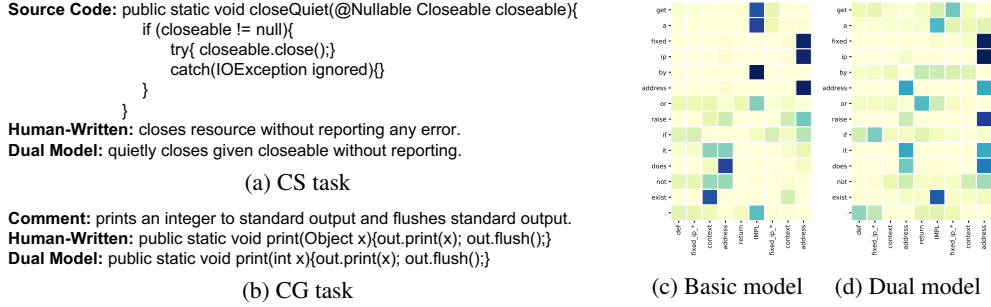

**Source Code:** public static void closeQuiet(@Nullable Closeable closeable){
    if (closeable != null){
        try{ closeable.close();}
        catch(IOException ignored){}
    }
  }
**Human-Written:** closes resource without reporting any error.
**Dual Model:** quietly closes given closeable without reporting.

(a) CS task

**Comment:** prints an integer to standard output and flushes standard output.
**Human-Written:** public static void print(Object x){out.print(x); out.flush();}
**Dual Model:** public static void print(int x){out.print(x); out.flush();}

(b) CG task

(c) Basic model    (d) Dual model

Figure 2: Qualitative analysis of our dual model. (a) Example of the generated comment given Java methods. (b) Example of the generated Java methods given natural language description. (c) Attention weights of basic model on CS task. (d) Attention weights of dual model on CS task.

value of the regularization term on attention is reduced from 10.51 to 4.757, indicating that the relation between the two models is enhanced after joint training.

**Qualitative Analysis and Visualization.** Figure 2a and Figure 2b show examples of CS task and CG task. We can see that the generated code and comment from the dual model have a very high semantic similarity with human-written. Figure 2c and Figure 2d show attention weights of a sample in the CS basic model and dual model. We can see that for words in the comment that are not aligned with the code, such as "a", the attention weights gained by basic model focus on a few specific words. On the other hand, the attention weights gained by the dual model are smoother, and therefore, we can get a better code representation for words that are not aligned. We think it is because the similarities between these tokens in the comment and the tokens in the code are different between CS and CG models. Since we add the attention constraint, the attention distributions of two models become close, making the attention weights of dual model smoother than them of the basic model in CS task.

**Discussion on Grammar Constraints.** To compare the dual model with the model that takes grammar rules as input for CG task, we evaluated SNM [Yin and Neubig, 2017] on Python dataset. SNM explicitly introduces the constraints of grammar rules when generating ASTs. The BLEU score for SNM is 8.095 and lower than our Basic model, indicating that the CG task on this dataset is very challenging. In particular, all prediction of SNM is valid, whereas the percentage of valid code generated by the dual model is low (Table 3). Hence, it is advantageous for the current dual model to constrain the generated code to satisfy the grammar rules, which will increase the percentage of valid code. Noting that the dual learning is a paradigm for joint training CS and CG. Integrating grammar rules into one model does not affect the dual relationship between the two models, and we leave it as our future work.

# 6 Conclusion

In this paper, we aim to build a framework which uses the CG as a dual task for the CS. To this end, we propose a dual learning framework to jointly train CG and CS models. In order to enhance the relationship between the two tasks in the joint training process, besides applying the constraint on probability, we creatively propose a constraint that exploits the nature of the attention mechanism. In order to confirm the effect of our model, we conduct experiments both on Java and on Python datasets. The experimental results show that after the dual training process, the CS model and the CG model can surpass the existing state-of-the-art methods on both datasets. In the future, we plan to consider more information to improve the performance of the joint model further, e.g., we would like to take the grammar rules as input to improve the performance of CG.

# 7 Acknowledgments

We thank all reviewers for their constructive comments, Fang Liu for discussion on manuscript. This research is supported by the National Key R&D Program under Grant No. 2018YFB1003904, and the National Natural Science Foundation of China under Grant Nos. 61832009, 61620106007 and 61751210.

## Footnotes

[2]https://pytorch.org/

[3]https://github.com/

[4]For papers that provide the source code, we directly reproduce their methods on two datasets. Otherwise, we rebuild their models with reference to the papers.

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
