[Reviews · NeurIPS 2019]

Reviewer 1



There idea of leveraging duality has been applied in other contexts, but never in the context of CS/CG. The related work seems to imply (but I wasn't 100% sure if this was the claim) that even in the context of these other domains, the dual constraint on attention weights was new. Is that correct? The idea in the paper is original and is well explained. The improvements over the baselines seem small but real. One aspect of the evaluation that I did not like is that it dismisses approaches that take additional inputs, such as grammars. If the addition of grammars leads to a substantial performance boost, then this work is comparing against a sub-standard baseline, and given that grammars already exist for all the languages tested, requiring a grammar does not seem like a significant extra burden on the user. Post rebuttal: So if I understand correctly from your rebuttal, you are making two arguments regarding the comparison with techniques that use grammars: a) your tool is already competitive with tools that use grammars (at least based on that one experiment reported in the rebuttal), and b) Your technique has a lot of room for improvement by using grammars, since a large fraction of what you produce is not valid code. I think these are important points to make in the paper, and adding the data from the rebuttal into the paper (or at least the supplement) would strengthen it considerably.

Reviewer 2



This paper proposes to train simultaneously two models one for code generation and one for code summarization. The main observation is that these two tasks are dual and they can be constrained when trained simultaneously to take advantage of the duality. The constraints are related to the probability correlation and the symmetry of the attention of the two models. The models are based on traditional Seq2Seq models. The probabilistic correlation of the two models is used to define a regularization term, while the symmetry of the two tasks are reflected in regularization terms based on the attention weights matrices. The paper compares this approach with state-of-the-art approaches for two different datasets (Java and Python) and show improvements in the 1-2% ranges for scores such as bleu, meteor and rouge. While the duality observation is interesting, the improvement in performance is limited. The ablation studies in the paper show that each constraint based on the two dualities brings a tad bit of performance improvement. The paper is well written and straightforward to follow (some suggestions for improvement below). =============== UPDATE: Dual-task training for CS/CG is neat and in the light of the clarifications in the authors' feedback, I'm increasing my score 6->7. Could you please include in the paper the following (from the authors' feedback): the discussion on grammars as additional input, the results on valid code, the clarification on how the language model is used, how the warm state for the two models is achieved, the diagram with the architecture. I think all these clarifications will increase the quality of the paper.

Reviewer 3



This paper presents an interesting approach of using the duality relationship between Code Summarization (CS) and Code Generation (CG) to improve the performance of a neural model on both tasks simultaneously. The main idea is to exploit the fact that the conditional probability of a comment given some source code, and the conditional probability of source code given a comment, are both related by their common joint probability. Moreover, since both the tasks of CS and CG use an attention-based seq2seq architecture, this paper also proposes to add an additional constraint that the two attention vectors have similar distributions, i.e. the attention weight of comment word i to source token j for the CS task is similar to the attention weights of the same pair for the CG task. The method is evaluated on two datasets of Java and Python programs/comment pairs and the dual training outperforms several baseline methods including the same architecture trained without dual constraints (basic model). Overall, I liked the idea of exploiting the dual relationship between the code summarization and code generation tasks. The proposed dual regularization terms relating to the factorization of conditional probability distributions and similarity of attention matrices are quite elegant. The experiment results also significantly improve the baseline approaches, and the ablation results show that both the duality constraints are useful. One thing that wasn’t clear to me was which parts of the dual relationship modeling were novel and which parts were taken from previous works. For example, Xia et al. [2017] proposed a supervised learning approach for imposing duality constraints and presented a similar probabilistic duality constraint (similar to Equation 8). The learning algorithm also seems similar except with the addition of the second regularization constraint. Is the only novel thing proposed in the paper is the dual regularization constraint corresponding to the similarity of attention vectors in Equation 9? In the experiments, is Basic Model the same as the seq2seq with attention model? In the DeepCom [Hu et al. 2018a] paper, the DeepCom model outperforms both Seq2Seq and Attention-based Seq2Seq models on summarization of Java methods. Can the authors present some insights on why the basic model might be outperforming the DeepCom model? It was also not clear whether for comparisons between different baseline models in Table 2, all the models have comparable number of trainable parameters? In the dual task, there are essentially twice the number of parameters, so it would be good to state how to compensate the baseline models with equal number of parameters. In section 4.1, the text states that the original Python dataset consists of 110M parallel samples and 160M code-only samples, and that the parallel corpus is used for evaluating the CS and CG tasks. But in Table 1, it seems there are much fewer samples (18.5k) for Python dataset. I was also wondering why not use the comments from the parallel corpus to pre-train the language model rather than using the language model for Java dataset. In section 4.2, it states that the best model is selected after 30 epochs in the experiments. Is this the case for the basic model or the dual model? Also, is the case for Java or Python dataset? Minor: page 1: Varies of → Various page 1: Specifically, z → Specifically page 1: studies before has → studies before have page 2: attracts lots of researchers to work on → (possibly something like) has attracted a lot of recent attention page 4: larger than it of → larger than that of

[Author Response · NeurIPS 2019]

Table 1: Percentage of valid code on CG task.

| Methods | Java | Python |
|---------|------|--------|
| SEQ2TREE | 22.6% | 13.9% |
| Basic | 19.6% | 17.5% |
| Dual | **27.4%** | **25.2%** |

Figure 1: The overall dual training framework.

We thank all reviewers for the comments and feedback.

**R1-Key Novelty.** The joint training framework is proposed for the first time in the field of program modeling in the
paper. Besides, we design a new dual regularization term on the attention mechanism that has never been proposed in
other machine learning fields. The probabilistic constraint of Eqn. 8 is similar to the previous work [Xia et al. 2017].

**R1-Grammar Constraints.** Due to limited time, we evaluated SNM [Yin and Neubig, 2017] on Python dataset.
SNM explicitly introduces the constraints of grammar rules when generating ASTs. The BLEU score for SNM is
10.62 and similar to our Basic model, indicating that the CG task on this dataset is very challenging. In particular,
all prediction of SNM is valid, whereas the percentage of valid code generated by the dual model is low (Table 1).
Hence, it is advantageous for the current dual model to constrain the generated code to satisfy the grammar rules, which
will increase the percentage of valid code. Noting that the dual learning is a paradigm for joint training CS and CG.
Integrating grammar rules into one model does not affect the dual relationship between the two models, and we leave it
as our future work. In the paper, we only focus on the use of duality between CS and CG. We will add the comparison
and discuss the future work on our dual model in the paper.

**R2-Evaluation Metric.** To evaluate how much of the generated code is valid, we calculate the percentage of code that
can be parsed into an AST, as shown in Table 1. Dual training can increase the percentage of valid code. 27.4% of the
predicted code of the dual model in Java is valid, and 25.2% of the prediction in Python is valid. We will add the details
in our paper.

**R2-Grammar Constraints.** Please read our reply to Reviewer #1 (Grammar Constraints).

**R2-Python Dataset.** The original data has about 110K parallel pairs (after duplicate example removal), and we
mistyped the number in our paper, which can be confirmed from Barone and Sennrich [2017]. Wan et al. [2018] used
the dataset in their experiments and published their code on GitHub. We processed the data according to their code.
Specifically, we limited the maximum length of code and comments to 100 and 50 respectively, and filtered out samples
that cannot be parsed into ASTs. Finally, we used the 55k of data to train CS and CG models (Table 1 in the paper).
This training data is not enough for training a language model for comments. We will highlight the details in the paper.

**R2-Dual Training Framework.** Our dual framework is shown in Figure 1. Language models are only used to calculate
marginal probabilities $\hat{P}(x)$ and $\hat{P}(y)$. Except for Tree2Seq and RL+Hybrid2Seq (They used Word2vec to pretrain
their token embeddings), the token embeddings for other models are randomly initialized. Warm-starting means that we
pretrained CS and CG models separately, then applied dual constraints to the two models for joint training. Pretraining
is common in previous work [Xia et al. 2017; Li et al. 2018], and we found it can speed up the convergence process of
joint training. Without pretraining, we can also get results beyond baselines. We will add the details in the paper.

**R4-Key Novelty.** Please read our reply to Reviewer #1 (Key Novelty).

**R4-Basic Model vs. Baselines.** We use a bidirectional LSTM (bi-LSTM) as the encoder and use beam search in the
inference process, which is different from the Attention-based Seq2Seq in [Hu et al. 2018a]. Besides, the attention
calculation in our model (Eqn. 2) is different from the method in [Hu et al. 2018a]. They did not report hyperparameters
of baselines in their paper; hence, for our basic model, we use the same hyperparameters as our dual model.

**R4-Parameters in All Models.** Since CS and CG models are trained at the same time and the parameters of the
two models are separate after the joint training, i.e., the two models solve their respective tasks separately after the
joint training, the number of parameters of each dual model is the same as that of the basic model. The number of
parameters for all models on Java CS task is as follows: CODE-NN 34M, DeepCom 53M (We use a bi-LSTM as the
encoder to ensure the number of parameters is comparable to that of our model.), Tree2Seq 52M, RL+Hybrid2Seq 80M,
API+CODE 80M (We set the embeddings and GRU states to 512 dimensions.), Basic model 53M and Dual model 53M.
The relationship between the number of all models' parameters is consistent in Java and Python experiments.

**R4-Python Dataset.** Please read our reply to Reviewer #2 (Python Dataset).

**R4-Training Details.** We select the best model according to the performance of the validation set after joint training
thirty epochs in both Java and Python experiments for our dual model. We did not use the test data to tune parameters.

[Meta-Review · NeurIPS 2019]

All reviewers liked the dual relationship between the code summarization and code generation tasks. They were also satisfied with the implementation and experiments. These problems are both difficult and important, hence progress is of interest even if such dualities have been identified in other contexts.